# SyGRID: Synthetically Generated Realistic Industrial Dataset

## Abstract

Industrial automation depends on accurate object recognition and localization tasks, such as depth estimation, instance segmentation, object detection, and 6D pose estimation. Despite significant advancements, numerous challenges persist, especially within industrial settings. To address these challenges, we propose SyGRID, (Synthetically Generated Realistic Industrial Dataset), a new simulated, realistic dataset specifically designed for industrial use cases. Its novelty lies in several aspects: the generated frames are photo-realistic images of objects commonly used in industrial settings, capturing their unique material properties; this includes reflection and refraction under varying environmental light conditions. Moreover, SyGRID includes multi-object and multi-instance cluttered scenes accurately accounting for rigid-body physics. Aiming to narrow the currently existing gap between research and industrial applications, we also provide an exhaustive study on different tasks: namely 2D detection, segmentation, depth estimation and 6D pose estimation. These tasks of computer vision are essential for the integration of robotic applications such as grasping. SyGRID can significantly contribute to industrial tasks, leading to more reliable robotic operations. By providing this dataset, we aim to accelerate advancements in robotic automation, facilitating the alignment of current progress in computer vision with the practical demands of industrial robotic applications.

## 1 Introduction

Industrial environments present unique challenges for 6D pose estimation and grasping due to the diverse appearances of objects. Accurately estimating the pose of such objects is crucial for tasks like robotic manipulation, part localization in assembly lines, and quality inspection. However, existing datasets often fail to adequately capture the complexity of industrial scenes, limiting the performance of pose estimation algorithms in real-world applications. This is due to different reasons. Recent progress on 6-DoF (degrees of freedom) pose estimation has made significant advancements Wang et al. (2020), He et al. (2020), Su et al. (2022), Kehl et al. (2017), Peng et al. (2019), Wang et al. (2021), Wen et al. (2023), but pose estimators are usually trained for a limited number of objects. Therefore, researchers in robotics are often unable to conduct real-world-based experiments leveraging 6D estimators trained on these datasets. Moreover, the annotation phase for thousands of real images is extremely time-consuming, as pose labelling tools exist, but they typically require on the order of minutes for a single object. In addition, the label accuracy could be affected by precision errors, due to human limitations during manual annotation. Some datasets use AprilTags or ARTags detection, as in Hinterstoisser et al. (2013), Chen et al. (2022a), Kaskman et al. (2019), Hodan et al. (2017), to annotate the pose, but using them during the training process could affect learning. Furthermore, datasets that do not require these markers are more adaptable and versatile to real-world applications. Moreover, some depth sensors may struggle with transparent or reflective objects. This means that real ground truth depth is not always correct, hence negatively impacting the learning process. For these reasons, many researchers moved to Physically-based Rendering (PBR) for automated dataset generation: synthetic data can be easily generated with low cost and high-efficiency thanks to modern simulators. However, a new problem arises: networks usually suffer from the domain gap between simulation and reality. This is true, especially for some scenarios that are common in the industry: aggregated objects, and reflective and transparent materials. To address this gap, we release a new simulated dataset tailored for multiple tasks in industrial settings,

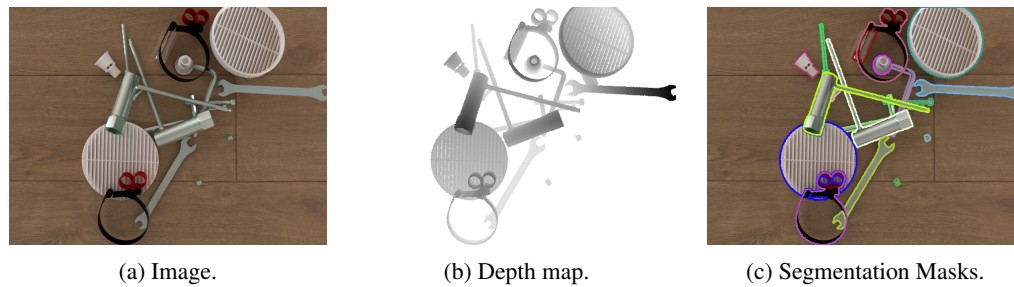

(a) Image.  (b) Depth map.  (c) Segmentation Masks.

Figure 1: One sample of the dataset, with depth and segmentation labels.

specifically for robotic manipulation. Our dataset includes a diverse set of objects and backgrounds commonly found in manufacturing environments. To the best of our knowledge, this is the first highly realistic simulated dataset including reflective and transparent objects with ground truth depth maps, instance masks, 2D and 3D bounding boxes and 6D pose estimation labels. The dataset is accessible to everyone, and the selected objects are commonly found in the industry. Ground truth labels are accurate because they are extracted from the simulation. By proposing this dataset, we aim to address different challenges of the 6D pose estimation domain Thalhammer et al. (2024):

- Occlusions and highly cluttered scenes;
- Different light conditions;
- Reflections and refraction given by objects' metallic or transparent materials.

The dataset is publicly available for download. We also evaluated it on different tasks, all related to the field of robotic manipulation: 2D bounding box detection, instance segmentation, depth estimation and instance-level 6D pose estimation. Section 2 focuses on existing datasets with multi-annotations (depth maps, poses, masks and bounding boxes), highlighting why our dataset represents a novelty from different points of view. In Section 3 we go deeper into details, by explaining how the data generator works and showing statistics on the released dataset. Finally, we trained multiple neural networks on different tasks, demonstrating in Section 4 remarkable results in both simulated test sets and on manually annotated real image frames. An overview of the dataset and labels is shown in Figure 1 and it is available at [1].

## 2 RELATED WORKS

In recent years, 6D pose estimation has achieved remarkable results. However, bridging the gap between research labs and real-world scenarios remains challenging, especially in industry, primarily because applications heavily depend on existing datasets. Commonly used State-Of-The-Art (SOTA) 6D pose datasets aim to generalize reality as much as possible, and use everyday common objects, while industries require specific scenarios designed exactly for their task. In our dataset, we focused on creating photo-realistic images in different industrial settings, trying to capture the real-world complexity of robotic scenarios for industrial bin-picking. Benchmark datasets like LM (Hinterstoisser et al., 2013), YCB-Video (Xiang et al., 2017), HOPE (Tyree et al., 2022), TUD-L (Hodan et al., 2018), and HB (Kaskman et al., 2019) present everyday scenes with objects unlikely to be seen in real-world industrial settings. In contrast, our focus is on the industrial domain, featuring objects commonly found in industry. Our dataset aims to present challenging simulated scenes where objects are occluded and cluttered, as in 6IMPOSE (Cao et al., 2023) and BlenderProc (Denninger et al., 2023), with unusual and more challenging materials, such as transparent or highly reflective metals without textures. The setting is similar to Gen4Industry, presented by Govi et al. (2024), however, some crucial elements differ: it is limited on single object per image and as a consequence, objects are not higly-cluttered. While some industrial datasets, such as Höfer et al. (2021), share similar goals to us like the highly cluttered environment, they are limited to one non-reflective object and this makes the task easier. T-LESS (Hodan et al., 2017) includes industry-typical objects and some occlusions, but the scenes are not cluttered, and the textureless objects do not present metallic or transparent

---

[1]https://mega.nz/folder/Km5nhThb#WJj68GLt_iNJipDG0gU2Xg

surfaces. ITODD (Drost et al., 2017) addresses challenges similar to ours, including reflective and metallic materials, but our dataset offers a wider range of scenarios. We varied background materials, object materials, as well as lighting conditions and camera distances, while maintaining a high level of realism. Other state-of-the-art datasets with reflective metallic textures, such as MP6D (Chen et al., 2022a) and ContourPose (He et al., 2023), include some occlusions but do not feature cluttered environments. Moreover, they are real datasets with ARUCO markers, which could influence learning. Recently, a significant advancement in the field of robotics-ready manipulable objects has been made by Handal (Guo et al., 2023). It is composed of real images acquired from multiple videos, with annotations for category-level object pose estimation and affordance prediction, and it focuses on hardware and kitchen tool objects. In addition to the fact that they did not focus on a specific industrial environment, our dataset differentiates from them for multiple reasons. First, our dataset is completely simulated, this poses a new challenge of high realism to fill the domain gap between simulation and reality. Secondly, we generate one single image with a set of parameters, by changing light conditions and backgrounds at each acquired frame and this enhances the variability, while frames of the same real video will present similar scenes. Thirdly, we provide the ground truth of cluttered multiple objects in a single image. Some recent works include transparent objects in their datasets, such as ClearPose (Chen et al., 2022b), PhoCal (Wang et al., 2022), KeyPose (Liu et al., 2020), and a domain-adapted dataset (Dai et al., 2022). However, these datasets are not specifically designed for industrial robotic picking applications, presenting everyday objects in occluded but not cluttered scenarios. The most similar dataset in terms of purpose and scenario is ROBI (Yang et al., 2021), which deals with highly reflective objects in robotic bin-picking applications, acquiring 8K images for 7 objects in the real world. However, ROBI features single-object multi-instance real images. Our dataset, on the other hand, is fully simulated, includes multi-object scenes, and offers varied lighting and background conditions to enhance generalizability across multiple situations. For these reasons, we believe our new dataset will significantly contribute to the field by addressing these gaps and providing a more comprehensive resource for industrial applications. Table 1 highlights differences between our dataset and existing ones, focusing on the variability of scenes. The comparison includes real and simulated datasets, also involving a further distinction between occlusions and highly cluttered scenes, which we are going to explain. While some datasets contain occluded objects, like LM-O or YCB-V, occlusions are given by the camera position, not by overlapping objects. The 'Highly cluttered' flag is the next level of occlusion, meaning that objects could be overlapped one on the other, as in ROBI or ITODD.

| Methods | Industr. scenario | Highly Cluttered | Type of images | Occl. | Reflective Metallic mat. | Transparent mat. | Multi objects | Multi instance |
|---|---|---|---|---|---|---|---|---|
| LM Hinterstoisser et al. (2013) | ✗ | ✗ | Real | ✓ | ✗ | ✗ | ✓ | ✗ |
| YCB-V (Xiang et al., 2017) | ✗ | ✗ | Real | ✓ | ✗ | ✗ | ✓ | ✗ |
| HOPE (Tyree et al., 2022) | ✗ | ✗ | Sim | ✗ | ✓ | ✗ | ✓ | ✗ |
| TUD-L (Hodan et al., 2018) | ✗ | ✗ | Real | ✓ | ✗ | ✓ | ✗ | ✗ |
| ITODD (Drost et al., 2017) | ✓ | ✓ | Sim | ✓ | ✓ | ✗ | ✓ | ✓ |
| T-LESS (Hodan et al., 2017) | ✓ | ✗ | Real | ✓ | ✗ | ✗ | ✓ | ✓ |
| MP6D Chen et al. (2022a) | ✓ | ✗ | Real | ✓ | ✓ | ✗ | ✓ | ✗ |
| ROBI (Yang et al., 2021) | ✓ | ✓ | Real | ✗ | ✓ | ✗ | ✗ | ✓ |
| ContourPose (He et al., 2023) | ✓ | ✗ | Real | ✓ | ✗ | ✓ | ✗ | ✓ |
| Handal (Guo et al., 2023) | ✓ | ✗ | Real | ✗ | ✓ | ✗ | ✓ | ✗ |
| Imitrob (Sedlar et al., 2023) | ✓ | ✗ | Real | ✓ | ✗ | ✗ | ✗ | ✗ |
| KeyPose (Liu et al., 2020) | ✗ | ✗ | Real | ✗ | ✗ | ✓ | ✗ | ✗ |
| DREDS | ✗ | ✗ | Sim | ✓ | ✗ | ✓ | ✓ | ✗ |
| ClearPose (Chen et al., 2022b) | ✗ | ✗ | Real | ✓ | ✗ | ✓ | ✓ | ✓ |
| TransGC (Fang et al., 2022) | ✗ | ✓ | Real | ✓ | ✗ | ✓ | ✓ | ✗ |
| PhoCal (Wang et al., 2022) | ✗ | ✗ | Real | ✓ | ✓ | ✓ | ✓ | ✗ |
| (Kleeberger et al., 2019) | ✓ | ✓ | Sim | ✓ | ✗ | ✗ | ✗ | ✓ |
| **Ours** | ✓ | ✓ | Sim | ✓ | ✓ | ✓ | ✓ | ✓ |

Table 1: Comparison between SyGRID and other datasets that provide labels for segmentation, 2D detection, depth and pose estimation. Details are in Section 2.

# 3 SYGRID

To support robotic manipulation research effectively, our dataset must meet several critical criteria:

| Dataset | PIQE ($\downarrow$) |
|---|---|
| SyGRID real | 2.91 |
| SyGRID syn | 13.34 |
| ITODD | 39.21 |
| YCBV (BlenderProch) | 14.71 |
| Gen4Industry | 31.94 |
| T-LESS | 24.64 |

Table 2: PIQE results, quality assessment.

| | |
|---|---|
| Number of objects | 10 |
| Min objects count | 1 |
| Max objects count | 20 |
| Mean Objects count | 15.51 |
| Number of frames | 10, 048 |
| Number of instances | 155, 842 |
| Resolution | $640 \times 480$ |

Table 3: Statistics, Part II.

- High realism. In industrial applications, where real data annotation is expensive, the generation of synthetic data is a compelling alternative. However, overcoming the domain shift between training and test data is still an open challenge, as claimed by Thalhammer et al. (2024). Therefore, preserving a high realism of the scenes becomes imperative in our work.

- Occlusion and 'interaction' among objects: they are not simply on a plane, they could also interact with each other by overlapping. This phenomenon could happen in our simulator and we define these kinds of scenes as 'Highly Cluttered', as previously explained in Section 2.

- Proper Size and Shape for Grasping: the objects should be designed or selected with shapes and sizes that can be manipulated by a wide range of robotic end effectors.

- Material properties variability: objects could be partially reflective, metallic, transparent, or opaque. Other datasets do not contain all these types of materials in one single scene, as shown in Table 1.

- Multi-object, multi-instance scene variations: as shown in Table 3, the number of objects and instances per scene is not predefined. On the contrary, the number of object types and instances is randomly chosen during the simulation to increase variability.

While qualitative visual analysis is important, quantitative assessment of objective realism and high image quality is fundamental. To this end, we computed two widely recognized metrics in this field: the Perception Image Quality Evaluator (PIQE) (Venkatanath et al., 2015) for quality assessment, and we innovatively utilized CLIP-IQA (Contrastive Language-Image Pre-training - Image Quality Assessment) for realism evaluation (Wang et al., 2023). An image is considered of high quality if the PIQE score is lower than 20 [2]. Specifically, we computed PIQE on different synthetic datasets, and Table 2 shows that our dataset achieves lower (better) PIQE scores than other simulated images, with camera-taken pictures achieving the lowest PIQE scores, indicating the highest image quality. We then computed CLIP-IQA in an unconventional way. CLIP-IQA is specifically designed for quality checks, where two textual prompts are given to an image; in the cited paper, these prompts correspond to "good photo" and "bad photo", and the model outputs the probabilities associated with each, providing a probabilistic measure of its quality. In our case, we adapted it for realism checking by using the prompts "Real Photograph" and "Generated Image" and running CLIP-IQA (with the "openai/clip-vit-base-patch32" pretrained model[3]) to obtain the probabilities for our real and synthetic images. The probability of "Real Photograph" for the real dataset is 0.99, while for the rendered images it is 0.81. Therefore, given that the model never misclassifies real photographs, the fact that it assigns an average probability of over $80\%$ to our rendered images being "Real Photograph" indicates that our synthetic images are highly realistic and could potentially be mistaken for real photographs. We highlight here that, independently from these promising results about quality and photorealism, the actual goal of this generated dataset is being able to train neural networks for use in real-world industrial applications; for this purpose, in Section 4 we provide an in-depth evaluation on tasks in the domains of artificial vision and robotics.

## 3.1 SET OF OBJECTS

The choice of the objects to be featured in our dataset accounts for their likelihood to be found in industrial settings. Also, for their peculiar features in terms of materials, shape and sizes, that would

---

[2]https://pypi.org/project/pypiqe/
[3]https://huggingface.co/openai/clip-vit-base-patch32

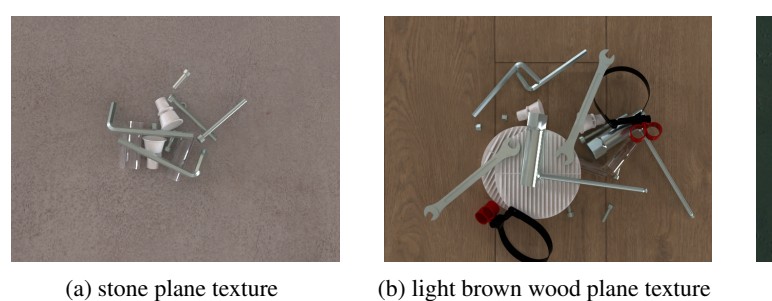

(a) stone plane texture       (b) light brown wood plane texture       (c) green metal plane texture

Figure 2: Examples of scenes with background and light variations.

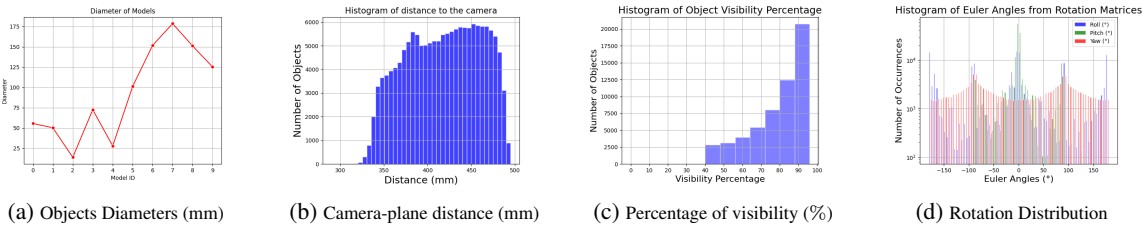

(a) Objects Diameters (mm)    (b) Camera-plane distance (mm)    (c) Percentage of visibility (%)    (d) Rotation Distribution

Figure 3: Statistics, Part I.

constitute an interesting test-bench for the chosen computer vision tasks. Our dataset contains 10 distinct objects. In Figure 4 each object is described by an image, its name and its material properties. Multiple objects and multiple instances of the same one could appear in a single image, where we define an object as the class or CAD model, and an instance as the occurrence and realization of that object in the image.

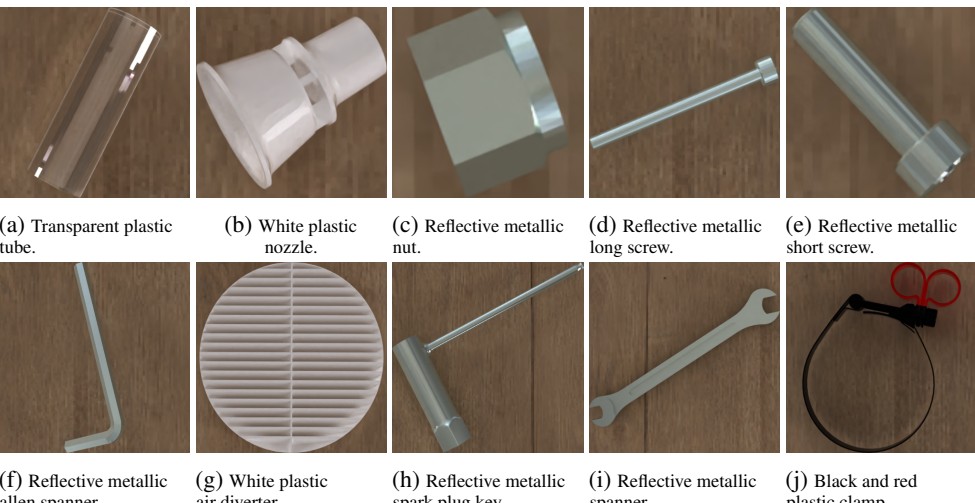

(a) Transparent plastic tube.    (b) White plastic nozzle.    (c) Reflective metallic nut.    (d) Reflective metallic long screw.    (e) Reflective metallic short screw.

(f) Reflective metallic allen spanner.    (g) White plastic air diverter.    (h) Reflective metallic spark plug key.    (i) Reflective metallic spanner.    (j) Black and red plastic clamp.

Figure 4: SyGRID Objects

## 3.2 DATA GENERATION AND RENDERING

The RGB images and data included in the presented dataset are synthesized through a stand-alone physically based rendering (PBR) application we developed and run on a Nvidia RTX 2080. Such a renderer is capable of accounting for global illumination, hence enabling us to model how light interacts with a variety of realistic materials, having diffusive, reflective, refractive and emissive components. Light is modelled as a ray travelling from the origin of the camera towards the scene (path tracing), therefore having a ray passing through each image pixel. When a ray hits the surface

of the material, the respective albedo is accumulated and a new ray is spawned with a direction that stocastically depends on the kind of material hit. This process continues until an emissive material is found, or a previously specified threshold of ray-light bounces is reached. The colour so far accumulated represents a radiance sample, as the rendering equation behind a PBR renderer is formulated as an integral which is impossible to solve in a closed form Kajiya (1986). For this reason, we approximate such a formula using a Monte Carlo estimator. In our implementation, a sample for each pixel is accumulated at every frame and each RGB image to synthesize is the result of combining a pre-determined amount of frames. Due to the stochastic nature of Monte Carlo estimation, the images can still contain noise hence indicating that some pixels did not fully converge to the target radiance described by the rendering equation. In order to remove noise, an AI-based denoising pass Áfra (2024) is applied to the final RGB image. The procedure described is fairly common in computer graphics literature and we redirect the interested reader who wants to know more about the algorithmic steps and the mathematical derivation of the Monte Carlo estimator for a PBR path tracer to the relevant well-known literature (Pharr et al., 2023).

Since our path tracer has been developed from scratch as a cross-platform OpenGL application that fully runs on the GPU, we have complete control on the implementation of the realistic set of materials in our industrial use case. This allows us to tailor the shaders (functions to be executed on the GPU with the aim to determine the colour of each image pixel) on the specific subset of materials featured by the set of objects we want to render.

In addition to the data needed for describing the scene to render, we extended the shaders so to have access to auxiliary buffers. These are regions of memory associated to each pixel in which we can store arbitrary data. In our specific use case, we store a tuple containing the object id, instance id and a depth value. The latter is trivially calculated as the pixel to object surface distance value. It only depends on the geometric and positional parameters of the hit objects, hence material properties such as translucency and reflections do not alter the precision of the extracted depth map. Auxiliary buffers are only written while rendering the first frame.

To simulate realistic cluttering and objects' positioning, we connected our path tracer with another application we developed with the aim of performing rigid body simulations [4]. In such an application, objects and their instances are spawned in random positions above a static plane so to allow them to fall, roll and clutter in a realistic manner. We then store the resulting 6D poses in the form of a 4x4 matrix to be used as input to our renderer.

From a given scene containing a number of cluttered objects, our renderer can be also configured to render each object individually within the same scene, hence "hiding" the remaining objects. These *image masks* allow us to calculate the visibility percentage of each object in relation to its degree of occlusion of the considered scene. This is crucial for establishing the visibility threshold used in labelling the dataset for computer vision tasks and for determining how well a network can learn based on the target object's visibility.

The renderer we implemented for the creation of this dataset is highly customizable, just as much as the recently proposed dataset generators, e.g. Greff et al. (2022); Singh et al. (2023). These dataset generators also exploit PyBullet for physics and PBR for global illumination, however they depend on Blender [5] for the rendering part. Which, in our opinion, make it less straightforward to use with respect to the standalone renderer we developed.

### 3.3 DYNAMIC PARAMETERS OF THE RENDERER

The scene creation process leverages the 6D position of the instances along with a selection of randomly varied parameters: light emission from the environment, camera position, starting position of objects, and plane texture. The specifics of how each parameter is set and modified for each scene are detailed below:

- **Background and Lighting**: In our renderer, the environment is represented by HDR (High Dynamic Range) environment maps, which provide information about both the background colour and the emissive light sources from the surrounding environment. These

---

[4] https://pybullet.org/
[5] https://www.blender.org/

maps influence the scene's lighting and are randomly selected from a set of 10 available equirectangular HDR textures sourced from polyhaven[6].

- **Camera Position**: The camera's position is not adjusted with each scene. Instead, it changes incrementally every $x$ scene, where $x$ is calculated by dividing the total number of experiments by the predetermined number of camera positions.

- **Starting Objects' position**: During rigid body simulations, a number of objects are initially positioned randomly to simulate natural interactions as they fall, roll, and accumulate. To vary their initial positions, the $x$ and $y$ coordinates of these objects are sampled from a uniform distribution between *-interval* and *interval*. The height from which these objects fall is parameterized as a function of their instance id. Later, during the rendering phase, the magnitude of this interval increases as the camera's distance from the plane increases, thereby minimizing the probability of an object to fall outside the viewable area of the frame.

- **Plane Texture**: To further add to variability, the textures used for providing the albedo of the plane in which the objects are collected during their fall is also a dynamic parameter that changes among rendered images. Once a texture is selected among a pre-defined set, its rotation around the z-axis is randomly set to one of the following angles: $0°$, $90°$, $180°$, or $270°$. Some examples of these different textures are provided in Figure 2.

## 4 EXPERIMENTS FOR BASELINE

To effectively support research in robotic manipulation, SyGRID must be validated for the learning tasks it was designed to address. To achieve this, we employed several state-of-the-art methods for tasks commonly encountered in computer vision and robotics. Specifically, we investigated four different types of deep models for four distinct tasks: object detection, instance segmentation, depth estimation, and 6D pose estimation. For this phase, we used a Nvidia RTX 4090 GPU.

### 4.1 2D DETECTION AND INSTANCE SEGMENTATION

In Section 3.2, we describe our method for obtaining precise and reliable segmentation masks for each object, despite the presence of occlusions. These masks enable straightforward derivation of 2D bounding boxes. To assess the realism of our synthetic datasets and their applicability in real-world scenarios, we trained a YOLOv8 (type L, with input size of 640, 43.7 million parameters) [7] model for 2D object detection and instance segmentation tasks. The train set comprises $80\%$ of the entire dataset. Evaluation of the model using the mean Average Precision (mAP) metric, with a threshold between 50 and $95\%$, indicates robust performance across various object sizes, as shown in Table 4. However, the model exhibits some limitations in detecting smaller objects, which are often subject to significant occlusion. This aspect is a strength of our dataset, encouraging the community to develop increasingly effective solutions for real-world industrial challenges, such as occlusions, self-occlusions, and symmetries. Another key aspect of our dataset is the high quality we achieved relative to industrial reference scenarios. To demonstrate its effectiveness, we validated the methods on a small batch of real images. This validation shows that our dataset is an effective tool for training deep learning methods capable of generalizing to real-world environments (Table 4). This validation was performed using a RealSense *D435i* camera to acquire real test images, manually annotated. The resulting metrics, along with a visual inspection of the predictions, confirmed the model's efficacy and highlighted areas for potential improvement (Table 4). In detail, we annotated more than 150 instances including five of the 10 objects: the tube, nut, nozzle, long screw and spark plug key. These five objects contain reflective, opaque and transparent materials.

### 4.2 DEPTH ESTIMATION

Our dataset offers precise and reliable depth maps, as they are generated using the renderer described in Sections 3.2 and 3.3. The accuracy of these depth maps is determined solely by floating-point precision, which we have set to half-float (16 bits). In contrast, depth maps from real-world datasets often exhibit varying levels of noise depending on the sensor used (which can vary in quality and

---

[6]https://polyhaven.com/hdris
[7]https://docs.ultralytics.com

Table 4: Metrics on 2D object detection and instance segmentation for simulated and real validation images. For the simulated images we tested all objects, while for the real dataset we collected images of only 5 objects.

| | Test on simulation | | Test on real | |
| | 2D Detection | Instance Segmentation | 2D Detection | Instance Segmentation |
| | mAP50-95(↑) | mAP50-95(↑) | mAP50-95(↑) | mAP50-95(↑) |
|---|---|---|---|---|
| All | 0.926 | 0.771 | 0.721 | 0.452 |
| Tube | 0.929 | 0.858 | 0.555 | 0.41 |
| Nut | 0.721 | 0.388 | 0.571 | 0.176 |
| Nozzle | 0.97 | 0.901 | 0.818 | 0.759 |
| Long Screw | 0.914 | 0.717 | 0.776 | 0.527 |
| Spark Plug Key | 0.986 | 0.889 | 0.883 | 0.386 |
| Short Screw | 0.841 | 0.637 | | |
| Allen Spanner | 0.931 | 0.749 | | |
| Air Diverter | 0.994 | 0.956 | | |
| Spanner | 0.982 | 0.868 | | |
| Clamp | 0.991 | 0.747 | | |

price). Additionally, variations in lighting with transparent or highly reflective objects can further propagate errors. Therefore, having a simulated dataset with precise ground truth during training can offer a significant advantage, as long as the dataset is realistic and mitigates the issue of domain gap between synthetic and real scenes. This is difficult, especially for challenging materials such as highly reflective or transparent objects. To test the realism of our existing dataset, and the improvement it could bring to real-world applications, we fine-tuned a neural network on it. The aim of this experiment is to give a baseline on the dataset for the specific task of monocular depth estimation. In detail, we fine-tuned and tested DepthAnything (Yang et al., 2024), a new and promising method, on $80\%$ of our simulated dataset. Then we validated the results on $20\%$ of the same dataset. Finally, we acquired real RGB images and depth maps with a RealSense *D435i* of the same objects and tested the training for real-world applications. The errors and metrics in Table 5 shows that SyGRID learned how to predict new simulated images and, in addition, it is able to generalize also to reality. When observing these metrics, we must consider that there is inevitably an error around $2\%$ at 2 m on the RealSense acquisition, therefore the ground truth on real images contains a small error. In addition, we visually observed that RealSense fails in those areas of the image affected by reflections.

Table 5: Metrics for depth estimation

| Method | Encoder | Training DS | Test on simulation | | Test on real | |
| | | | AbsRel (↓) | $\delta_1$ (↑) | AbsRel (↓) | $\delta_1$(↑) |
|---|---|---|---|---|---|---|
| Depth Anything | ViTL | SyGRID | 0.00852 | 0.99998 | 0.09592 | 0.99845 |

## 4.3 6D POSE ESTIMATION

In robotic grasping, object pose recognition is becoming increasingly important, attracting growing interest from the scientific community in recent years. For this reason, it was imperative to evaluate the effectiveness of SyGRID with a network trained specifically for instance-level 6D pose estimation. Therefore, we selected the RGB-based GDR(Geometry-Guided Direct Regression)-Net (Wang et al., 2021), which underpins the recent winner of the BOP (Benchmark on Object Pose Estimation) challenge. We trained the network on the $80\%$ of the dataset. Despite its reliance solely on RGB data, GDR-Net achieves commendable performance across most object classes, though it encounters difficulties with smaller items such as nuts and short screws. This limitation is probably due to the small number of pixels that describe these kinds of objects and the resize applied to the cropped images, which could increase noise. Additionally, we conducted visual assessments using predictions generated on real images captured with a RealSense *D435i* camera. These visualizations confirm the network's relative accuracy in practical scenarios, underscoring the high degree of realism our dataset offers. This capability is crucial for enhancing the applicability of synthetic datasets in real-world 6D pose estimation tasks.

Table 6: Metrics on 6D pose estimation for simulated and real validation images. For the simulated images we tested all objects, while for the real dataset we collected images of only 5 objects.

| | Test on simulation 6D Pose estimation | | | Test on real 6D Pose estimation | | |
|---|---|---|---|---|---|---|
| | Re($\downarrow$) | Te($\downarrow$) | add-10($\uparrow$) | Re($\downarrow$) | Te($\downarrow$) | add-10($\uparrow$) |
| All | 14.11 | 0.01 | 77.65 | 19.06 | 0.03 | 47.24 |
| Tube | 18.56 | 0.01 | 85.69 | 41.58 | 0.01 | 54.55 |
| Nut | 2.47 | 0.01 | 16.90 | 14.62 | 0.07 | 0.00 |
| Nozzle | 1.00 | 0.00 | 93.22 | 6.77 | 0.01 | 80.49 |
| Long Screw | 1.88 | 0.01 | 90.37 | 11.90 | 0.02 | 57.14 |
| Spark Plug Key | 5.85 | 0.01 | 93.51 | 20.43 | 0.02 | 44.00 |
| Short Screw | 1.88 | 0.01 | 48.29 | | | |
| Allen Spanner | 6.22 | 0.01 | 83.37 | | | |
| Air Diverter | 2.97 | 0.00 | 99.15 | | | |
| Spanner | 87.34 | 0.01 | 81.29 | | | |
| Clamp | 12.91 | 0.03 | 84.66 | | | |

## 5 ROBOTIC APPLICATION

Our robotics pipeline is primarily designed to tackle the pick-and-place task, a fundamental operation in automation and robotics. This task involves identifying, grasping, and relocating objects from one point to another. The system is built to function seamlessly in various environments, particularly in industrial settings where repetitive yet precise actions are critical. In our scenario, we use the same specific objects used in SyGRID, described in Section 3. The core components of this pipeline are:

- **Robotic Arm**: We employ a Universal Robots UR5e, a collaborative robotic arm equipped with a two-finger gripper. The gripper can either be the Robotiq Hand-e or the OnRobot RG2, depending on our task's specific needs.

- **Visual Sensor**: The robotic arm is guided by a Realsense D435i camera, which acts as the system's visual input. The camera captures RGB images of the scene, which are then processed to identify and localize objects.

- **Object Manipulation**: After analyzing the images, the arm manipulates the objects by picking them up from the workspace and placing them into a designated container. Deep vision methods are essential for effective manipulation of objects in unstructured environments.

The entire system is built using ROS2 (Robot Operating System version 2), a widely used framework in robotics. ROS2 adopts a node-based structure, where each component or task is encapsulated in an independent node. These nodes communicate with one another by publishing and subscribing to specific topics, ensuring seamless data flow between the components. We have designed the system to leverage distinct conceptual nodes, allowing for smooth integration of various tasks like vision processing, object detection, and control of the robotic arm.

SyGRID plays a critical role in the training and validation of vision deep learning methods, shown in Section 4. Each method, whether it involves object detection, instance segmentation, 6D pose estimation, or depth estimation, is encapsulated within its own independent node. These nodes can either function individually or work in combination with others, creating flexible and adaptable pipelines. For real-world cluttered environments, our current best-performing solution combines Instance Segmentation, implemented with YOLOv8 for identifying and segmenting objects in the scene, and 6D Pose Estimation, using GDRN to estimate the position and orientation of objects. The results are shown in Figure 5 and in the video shared at this link [8].

## 6 CONCLUSION, LIMITATIONS AND FUTURE WORK

We presented SyGRID, a photo-realistic simulated dataset tailored towards applications within industrial settings. SyGRID features highly heterogeneous scenes, including diverse backgrounds and

---

[8]`https://mega.nz/file/a7QlwToQ#4UR2-ZS6hxKM6VmLNOpsqlJoUJd4znJZA4dgGtWcpLU`

Figure 5: Intermediate results published from the ROS2 nodes for instance segmentation (left), detection (centre) and 6D pose estimation(right).

ambient lighting, with specific emphasis on peculiar objects' material properties such as reflections and refractions that are known to have significant negative effects on common computer vision methods. Moreover, we have shown the application of SyGRID on four computer vision tasks hence demonstrating that our proposed dataset is able to significantly decrease the gap between synthetic images and camera-taken pictures. We highlight that the SyGRID dataset provides highly accurate, noise-free ground truth labels, without the need of *tags*, and overcoming the intrinsic level of noise that is typical of acquisition sensors. As for future work, we are planning to address on the limitations of this dataset. More specifically, while in the proposed dataset the distance between the camera and the plane is one of the many variables when rendering a scene, the angle in which the camera is looking at the scene is always perpendicular: this is a design choice driven by our experience on existing robotic arm pickers; however, it is not unthinkable that different industrial settings might exist with diverse camera inclinations with respect to the picking plane. Hence, in the future, we plan to add a rotation matrix to be applied to the camera extrinsic parameters. We also plan to further investigate *visibility threshold*, i.e. understanding for each specific application, what is the minimum threshold of visibility percentage that would allow a network to correctly perform its task as a function of the object's geometry, image resolution and the task at hand. On the rendering part, we plan to understand how our proposed approach and pipeline of operations would generalize to different scenarios other than the industrial domain. We also plan to investigate how to generate these kinds of datasets by editing images synthesized through novel rendering methodologies such as Neural Radiance Fields (NeRFs, (Mildenhall et al., 2021; Müller et al., 2022)) and Gaussian splatting (Kerbl et al., 2023), hence being able to populate annotated datasets using scene variations starting from camera-taken videos.

## REPRODUCIBILITY STATEMENT

SyGRID, real test data and models' weights are available at the dataset link in Section 1.

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
