# SyGRID: Synthetically Generated Realistic Industrial Dataset

## Supplementary Materials

## 1 Renderer Implementation and Generation Details

In Fig. 1 we show a schematized version of the pipeline of operations needed to generate annotated data to be then stored in the dataset.

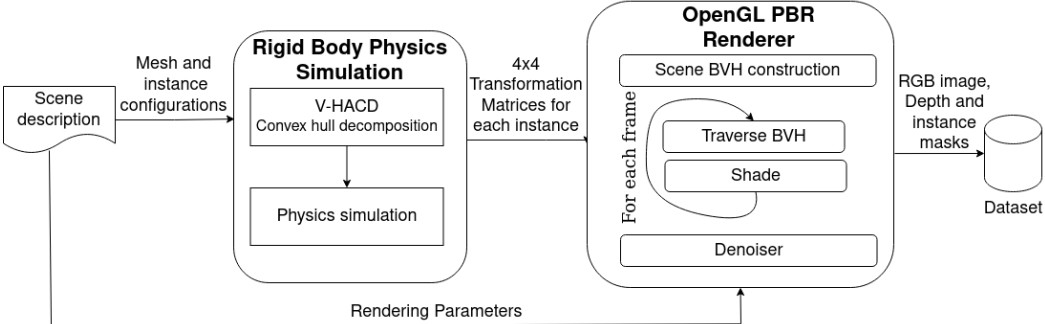

Figure 1: Pipeline of operation for generating a single image and respective labels.

The scene description is a *json* file containing the necessary parameters for both rigid body interaction and rendering simulations. It marks the initial phase of the pipeline, detailing the objects involved (file paths to 3D models). For the subsequent rigid body simulation phase, only mesh geometry information is necessary; the number of instances per object is specified as an additional input to the physics block. These are variables subject to change for each data item to be featured in the output dataset.

As far as rendering parameters are concerned, the same scene description file also specifies camera parameters and environment map. These are variables for data item to generate. In the same file, we also specify a number of fixed parameters that we did not change during data generation, namely the total number of frames (samples) needed to generate an RGB image (fixed to 5000), image resolution (fixed at 640x480), and the number of maximum ray-light interactions (path tracing depth, fixed to 32).

In the rigid body physics simulation phase, the V-HACD[1] algorithm is employed to decompose complex, concave objects into simpler convex hulls. This decomposition is crucial for performing efficient and accurate physics simulations. Following this, the pybullet[2] library is used to calculate the interactions, movements, and collisions of these objects based on physical laws, hence, outputting transformation matrices for each instance. These 4x4 matrices encode the position, rotation transformations necessary to accurately position each object instance within the 3D scene. Those matrices are stored in another file, that is going to be read (together with the scene description) by the rendering process.

The rendering process begins with the construction of a Bounding Volume Hierarchy (BVH) to expedite rendering. The BVH groups objects hierarchically, facilitating efficient traversal and shading.

---

[1] https://github.com/kmammou/v-hacd
[2] https://pybullet.org/

For each frame, the renderer traverses the BVH to determine which objects are visible from the camera's perspective and need to be shaded. Shading involves computing the final colour of each pixel using Physically Based Rendering (PBR) techniques. A denoiser is then applied to the shaded image to reduce any noise introduced during shading, enhancing image quality.

The renderer outputs an RGB image, its depth map, and instance masks. This data item is stored in the dataset. With our system configuration (Nvidia RTX 2080, Intel-Core I7-7700k), a data item is generated in less than 10 seconds on average, depending on the number and kind of instances.

## 2 REAL ANNOTATED DATASET

To fairly test the effectiveness of our dataset in the real world we annotated real, camera-taken pictures. Then, we evaluate those pictures with different networks, as trained on the proposed SyGRID dataset. In this Section, we detail the annotation procedures and highlight their limitations. The labelling phase requires significant effort, hence we reassert the advantage of using only-simulated datasets and the importance of closing domain-gap between reality and simulation.

### 2.1 INSTANCE SEGMENTATION AND 2D ANNOTATIONS

We used a tool ([3]) to manually annotate polygons of the objects. The tool is based on Segment Anything **?** and it makes it possible to edit the obtained polygons. After labelling, we converted labels into the YOLO format, repairing the broken polygons of single masks by connecting them. Visual

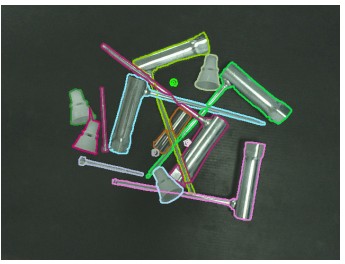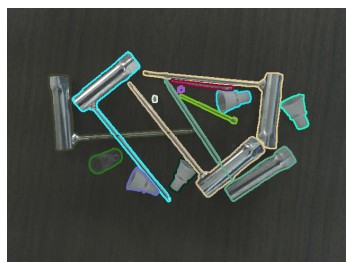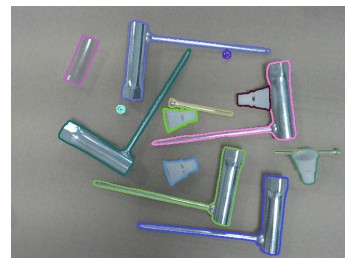

Figure 2: Real Images manually annotated with Segment Anything.

results show annotations on real images. Given the segmentation polygon we can easily compute the instance masks, hence obtaining 2D bounding boxes.

### 2.2 DEPTH GROUND TRUTH

Depth maps were acquired through a RealSense 435i, as explained in the paper. Then the predicted and ground truth depth maps on real images are compared. Observing the metrics on reality, we must consider that RealSense suffers of an error around 2%, as claimed by the manufacturer ([4]). In addition, we observed larger errors in highly reflective surfaces.

Recently, some more powerful and accurate depth sensors exist, but they are much more expensive.

On the other side, our dataset generator provides accurate ground truth depths at no cost and it allows us to train a network and predict more accurate depth maps. This is particularly useful in industrial environments, where settings are often repetitive and reproducible with our images. Therefore it is not necessary for the network to have generalization capabilities to move from one scenario to another. It is sufficient to create the appropriate dataset for that specific scene and train a network on it.

Furthermore, we used depth maps only as ground truth labels, but it would be possible to use them also as input for a learning process. In this case, to generalize to the real domain, a different level of noise, depending on depth sensor used, should be applied to the ground truth images.

---

[3] https://github.com/haochenheheda/segment-anything-annotator
[4] https://www.intelrealsense.com/depth-camera-d435i/

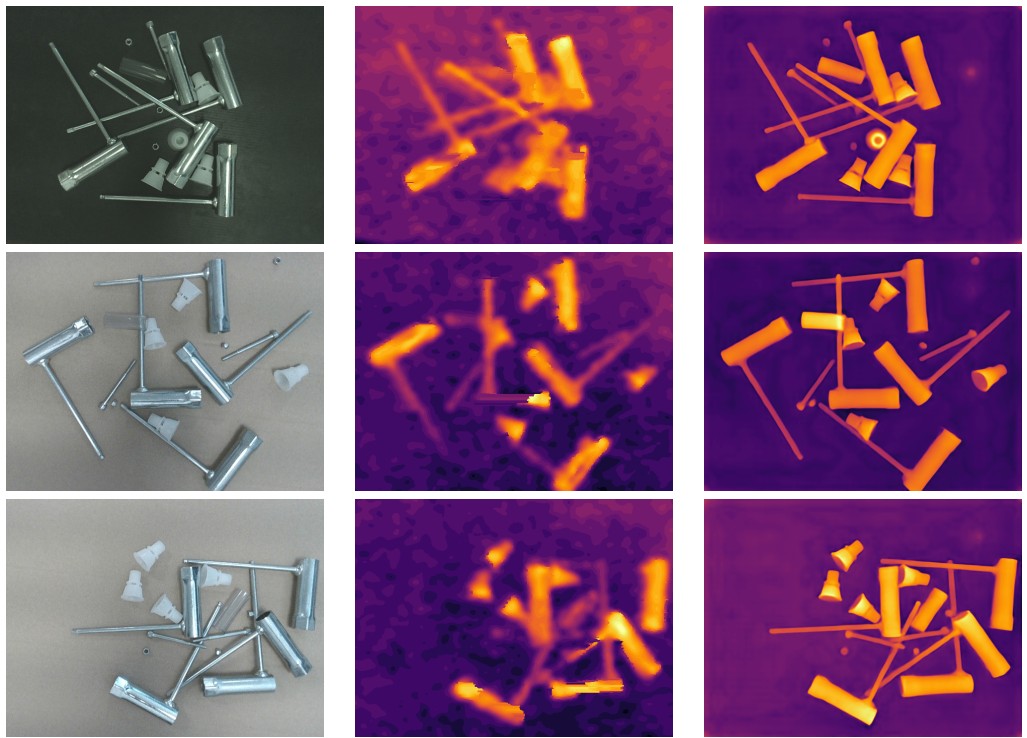

Figure 3: On the left: real RGB images. On the centre: depth maps on RGB acquired by RealSense, there are some errors given by reflections or transparencies. On the right: predictions of DepthAnything **?** based on the simulated dataset training.

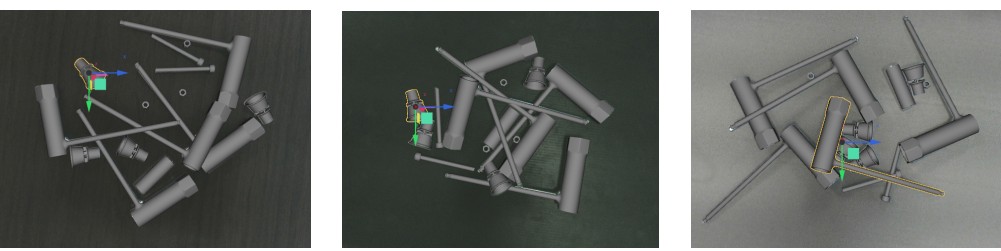

Figure 4: Examples of 6d cad models on the image with the tool 6d-pat, for 6D pose annotation.

### 2.3   6D POSE ANNOTATION

To annotate the 6D pose labels, we follow an autolabelling procedure. First, we take the prediction from our trained network, GDR-Net, and visualize the object CAD model on the real image. After that, we edit the position directly on the image by moving the CAD model as shown in Figure 4. To do this, we used a tool called 6D-PAT [5]. This annotation phase is extremely time-consuming. Despite this, the accuracy is not guaranteed, especially in the case of small objects, like the nut.

## 3   METRICS FOR EVALUATION

For 2D detection and instance segmentation, we used the well-known mAP (mean Average Precision), with a threshold between $50$ and $95\%$, as written in the paper.
We use the Absolute Relative Error and the $\delta_1$ for depth metrics. The first is defined as the average value over all the image pixels of the $L_1$ distance between estimated and ground truth depth (as

---

[5] https://github.com/florianblume/6d-pat

explained in **?**). It is scaled.

$$AbsRel = \frac{1}{N} \sum_{i=1}^{N} \frac{|D_i^* - D_i|^2}{D_i}$$

The $\delta_1$ is expressed as:

$$\delta_1 = \frac{1}{N} \sum_{i=1}^{N} (\max(D_i^*/D_i, D_i/D_i^*) < 1.25)$$

where $D^*$ is the ground truth depth and $D$ is the estimated depth. $N$ is the total number of pixels and $D_i$ and $D_i^*$ represent the estimated and ground truth depth values at the pixel indexed by $i$.

For 6D pose estimation we used ADD as metric (Average Distance of Model Points). Given the CAD model $\mathcal{M}$, the estimated rotation $\hat{\mathbf{R}}$ and translation $\hat{\mathbf{t}}$ and the ground-truth rotation $\overline{\mathbf{R}}$ and translation $\overline{\mathbf{t}}$ we define the error:

$$e_{ADD} = avg_{x \in \mathcal{M}} \|(\hat{\mathbf{R}}\mathbf{x} + \hat{\mathbf{t}}) - (\overline{\mathbf{R}}\mathbf{x} + \overline{\mathbf{t}})\|$$

If the model $\mathcal{M}$ has simmetries, then

$$e_{ADDs} = avg_{\mathbf{x_1} \in \mathcal{M}} \min_{\mathbf{x_2} \in \mathcal{M}} \|(\hat{\mathbf{R}}\mathbf{x_1} + \hat{\mathbf{t}}) - (\overline{\mathbf{R}}\mathbf{x_2} + \overline{\mathbf{t}})\|$$

The estimated pose is considered correct if

$$e < \theta_{ADD} = k_m d$$

where:

- $k_m$ is a constant generally equal to $0.1$,
- $d$ is the object diameter.

The rotational error $re$ between the estimated rotation matrix $\hat{\mathbf{R}}$ and the ground truth rotation matrix $\overline{\mathbf{R}}$ can be computed as follows:

$$re = \arccos\left(\frac{\text{trace}(\hat{\mathbf{R}}\overline{\mathbf{R}}\mathbf{x}^T) - 1}{2}\right)$$

where:

- $\text{trace}(\cdot)$ denotes the trace of a matrix, which is the sum of the diagonal elements.

The translational error $te$ between the estimated translation vector $\hat{\mathbf{t}}$ and the ground truth translation vector $\overline{\mathbf{t}}$ can be computed as follows:

$$te = \|\overline{\mathbf{t}} - \hat{\mathbf{t}}\|$$

where $\|\cdot\|$ denotes the Euclidean norm, measuring the distance between two points in 3-dimensional space.

## 4  IMAGE GALLERY

In Table 1 we present 10 different examples of the RGB images generated by our dataset, as produced by the renderer described in the paper.

## 5  REAL-WORLD DEMONSTRATION

To demonstrate the effectiveness of our dataset, we implemented the pipeline composed of YOLOv8 and GDR-Net on a real cobot (Collaborative Robot). In detail, we used a Universal Robot UR5e [6] and a RealSense 435i. We shared a video with a demonstration of picking with our pipeline [7].

---

[6]https://www.universal-robots.com/it/prodotti/robot-ur5/
[7]https://mega.nz/file/a7QlwToQ#4UR2-ZS6hxKM6VmLNOpsqlJoUJd4znJZA4dgGtWcpLU

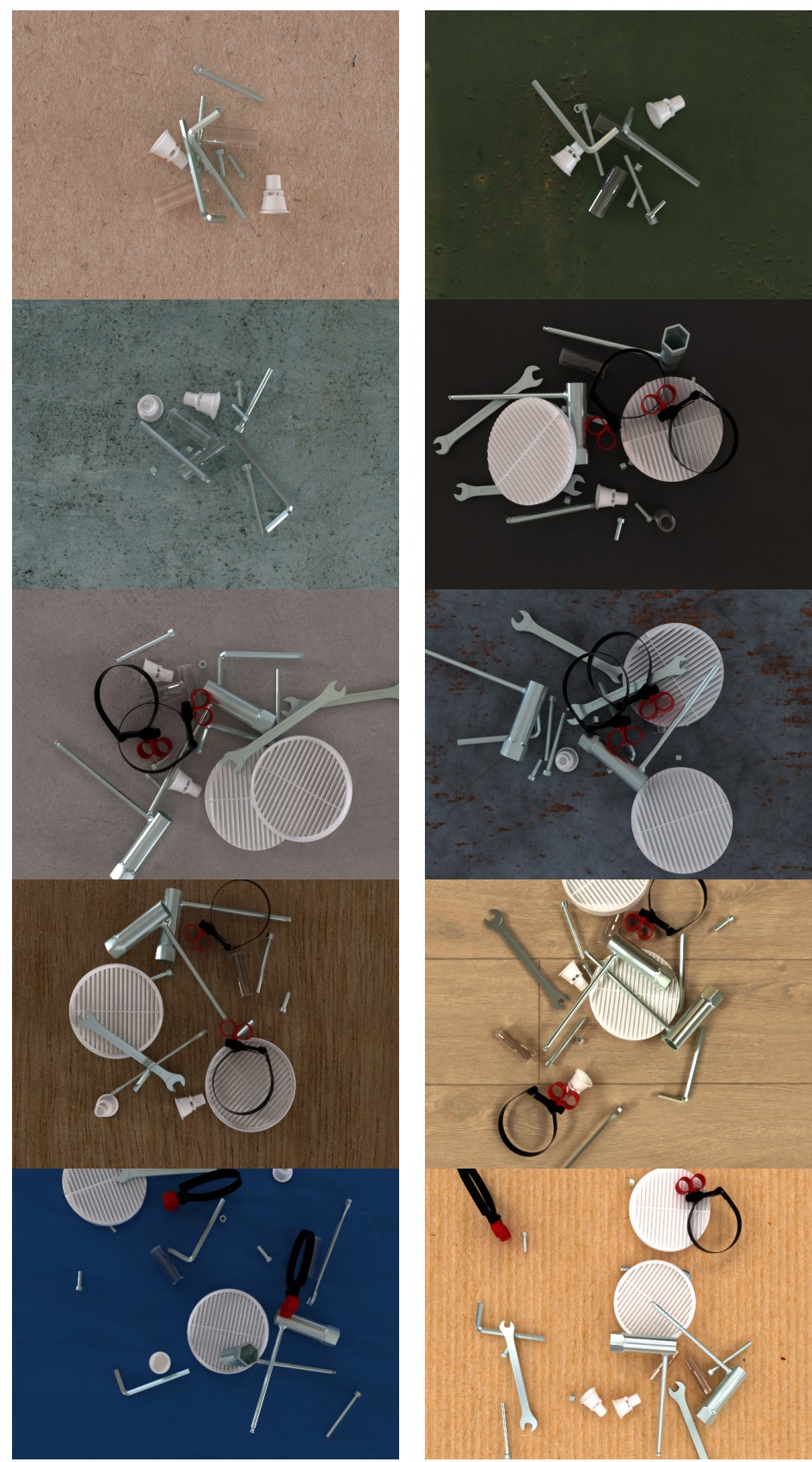

Table 1: Example RGB images produced for the dataset

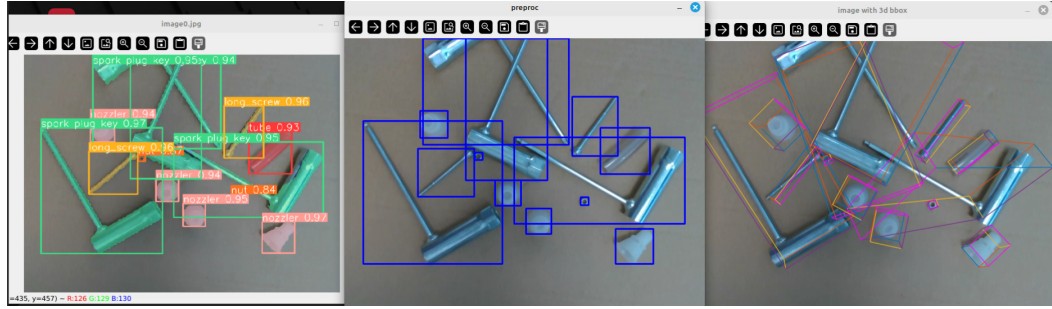

Figure 5: Inference of our pipeline on a real-world experiment of picking. On the left: YOLOv8 predictions. On the centre: bounding boxes given by the predicted masks. On the right: 3D bounding box based on the pose prediction from GDR-Net.