# OpenReview forum: "SyGRID: Synthetically Generated Realistic Industrial Dataset"
_ICLR.cc/2025/Conference — ICLR 2025 Conference Withdrawn Submission_

### Official Review · Reviewer_DiGx · 2024-10-31

**Soundness:** 2
**Presentation:** 2
**Contribution:** 2
**Rating:** 3
**Confidence:** 3

**Summary:**

This paper introduces SyGRID, a novel synthetically generated industrial dataset designed to enhance object recognition and localization tasks critical for industrial automation, such as 2D detection, segmentation, depth estimation, and 6D pose estimation. SyGRID’s key contributions include highly realistic, photo-realistic images that simulate industrial environments with challenging elements like material reflections, refractions, and cluttered multi-object scenes modeled with rigid-body physics. By offering a comprehensive dataset tailored for realistic industrial conditions, SyGRID bridges the gap between research and practical applications, enabling advancements in robotic tasks such as grasping, thereby supporting more robust robotic automation in industrial settings.

**Strengths:**

Realistic industrial dataset is needed for downstream robotic tasks, while this scenario is yet to be well studied and has many challenges, such as the cluttering and transparency of objects. I believe this dataset is practical and will contribute to the development of this field.

**Weaknesses:**

For presentation, I recommend authors to divide long paragraphs into some short paragraphs, which could largely increase the ease of reading of this paper.

For experiments, as the contribution and final goal is to facilitate downstream robotic manipulation, while this dataset is rendered in simulation, I recommend authors to demonstrate both qualitatively and quantitatively how much this simulation rendered dataset can bridge the sim2real gap of robotic perception and manipulation.

For dataset, are there diverse objects in a category? Or the diversity only exists in number of categories?

The real-world robotic manipulation video could be polished if the author presents the policy visualization, instead of only the robotic execution.

**Questions:**

Please see weakness.

---

> ### Author Response · Authors · 2024-11-18
>
> For presentation: thank you for your suggestions, we can improve it.
>
> About robotic perception, we demonstrate both qualitatively (image results of inference) and quantitatively (all table metrics on real images) how it works. About manipulation  experiments, we only show qualitative experiments because the quantitative experiments (successful robotic grasping percentage) are not reproducible: they are quite dependent on the robotic arm, the gripper, the position of objects, the lights.
>
> The objects' diversity exists in a number of categories. As written in the paper, we focused on instance-level 6d pose estimation. Our future purpose is to extend the work to category level 6d pose estimation.
>
> Thank you for the last request, a new video of the inference phase has been added to the SyGRID folder. You can find it at this link: https://mega.nz/folder/Km5nhThb#WJj68GLt_iNJipDG0gU2Xg/file/2zhXXDjZ
> In the video, the inference phase is recorded, and from left to right respectively you can find: instance segmentation, bounding boxes, 3D bounding boxes computed from the pose. The procedure is slow because you have to consider that in the meanwhile the robotic arm is moving and picking objects. For this reason, each step contains less objects than the previous one.

---

### Official Review · Reviewer_L67o · 2024-10-31

**Soundness:** 2
**Presentation:** 2
**Contribution:** 1
**Rating:** 3
**Confidence:** 4

**Summary:**

The paper introduces a new, synthetic dataset tailored for computer vision tasks, such as 2d object detection, segmentation, depth estimation, and 6DOF pose estimation in industrial settings. The dataset aims to bridge the gap between research datasets and real-world industrial environments by providing high-quality, photo-realistic data with challenging features such as reflections, refractions, and highly cluttered multi-object scenes. The dataset is evaluated on real and synthetic data using 2D bounding box detection, instance segmentation, depth estimation, and instance-level 6D pose estimation.

**Strengths:**

Overall, the paper is clearly written and easy to follow. It introduces prior work on industrial datasets, their strengths and weaknesses, and presents the varying attributes visually through tables. The paper further shows that the presented dataset is more realistic overall compared to prior proposed methods (about on par to YCBV) and, therefore, partially closes the sym-to-real gap.

**Weaknesses:**

I have multiple questions regarding the proposed dataset's motivation, technical contribution, and real-world usefulness.
1. To me, the dataset seems to be ill-motivated. The dataset focuses solely on applying industrial bin-picking with a very specific (and small) number of objects, severely limiting the applicability of the dataset to real-world scenarios. I am not aware of any application where these exact parts must be picked from an unorganized bin by a robotic manipulator. If such scenarios exist, they should be mentioned in the introduction. If the claim is that the dataset can be used for other applications, I would expect some experiments that analyze the generalization capability of algorithms trained on this dataset in real-world applications. I suggest that the paper contains a more thorough motivation of applications in industry where this dataset can be useful.
2. The motivation for introducing this dataset is to handle "occlusions and highly cluttered scenes" and "different light conditions." I would argue that industrial settings are some of the most controllable environments for robots. Not only can the lighting be fully controlled, but I also have trouble envisioning specific tasks where the robot has to pick these specific arrangements of objects.
3. Regarding the results, the paper claims that the main goal is to reduce the sym-to-real gap in performance. However, Table 2 shows that a large gap still exists between the two data distributions. Similarly, the results show that using this data for training results in widely differing performance on real vs synthetic test data (-15 to -20% on segmentation, 10x difference on depth estimation, between 2-6x large rotation error). These differences in performance show to me that the dataset is only partially useful to generalize to real-world data and does not solve the sym-to-real problem.
4. It is also questionable how a model trained on this dataset performs on out-of-distribution data. Can these models be used on some of the other datasets (Table 1.)? If so, what is the performance?

Overall, I do not believe that the paper contains enough motivation or technical novelty to justify publication at ICLR. The dataset is too generic in the tasks it addresses without providing real insights into real-world robotics manipulation challenges. I therefore recommend rejection of the paper.

**Questions:**

See above.

---

> ### Author Response · Authors · 2024-11-19
>
> 1-2. We understand your perplexities: industrial scenarios usually have fixed lights, and fixed objects and they are located at a precise point, in such a way that the robotic arm can successfully pick them. This is the point: it is a huge limitation in industries. Only the intelligent ability of grasping from humans is able to pick the objects from the aggregated scene and put it in the correct point for the robot to be grasped. In addition, if the object is unfortunately moved unexpectedly, the entire line stops, resulting in a significant loss of productivity. But at the moment, technologies are not mature enough to substitute these technologies with more intelligent ones. On the other hand, research is making notable progress in this field. One of the main obstacles is to bring these technologies to specific scenarios related to the needs of the industries. This dataset itself comes from the requests of multiple industries. Objects are currently used in these industries every day. The advantage of using a generated dataset is that in a few hours, we can generate a dataset specifically designed for a use-case and train a pipeline with no limitations, creating an intelligent robotic arm. In addition, fixing the lighting conditions could pose a limitation. For instance, unmanned ground vehicles (UGVs) equipped with robotic arms often operate in dynamic industrial environments, navigating heterogeneous settings where lighting conditions frequently change. In such applications, which are expected to become increasingly common in the industry, relying on fixed lighting solutions is simply not viable.
> 3. The performance gap observed between the deep model tested on synthetic data and real-world data can be attributed to several factors, primarily related to the quality and characteristics of the real dataset. First, the sensor used for real data acquisition (Intel RealSense) is a mid-to-low-end device, which inherently limits the quality of the captured data. Specifically, the depth data provided by RealSense is known to be noisy, as shown in supplementary materials. In contrast, the synthetic dataset provides precise and clean depth information, and the model is trained on them. Moreover, the RGB data captured by RealSense is also of relatively low quality, which significantly impacts methods relying on visual information (e.g., YOLOv8 and GDRNet).
> Second, the real dataset is inherently more limited in scope compared to the synthetic dataset. While the synthetic dataset is designed to encompass a wide range of scenarios, the real dataset captures only a small subset of these cases. This limitation translates to a narrower range of tested backgrounds, lighting conditions, and camera heights relative to the plane. It also results in a constrained variety of colors and reflections on metallic and transparent objects, which are critical features for the model. Consequently, the real dataset’s distribution is inherently different from that of the synthetic dataset.
> It is also important to acknowledge the limitations of the model itself. Models trained exclusively on synthetic datasets inherently struggle to generalize to real-world data due to the domain gap between synthetic and real distributions. While improving the quality and diversity of the real dataset would likely reduce the performance gap, the syn-to-real transfer remains a challenge, as models trained on synthetic data are not fully equipped to handle the complexities of real-world data. Finally, we would like to clarify that the choice to use the RealSense sensor was intentional. We believe that deep learning methods have the potential to compensate for sensor limitations in the future. By deliberately using a mid-tier sensor, we introduced an additional challenge in our work, aiming to highlight the robustness of the proposed approach and the importance of addressing these real-world constraints. In summary, while the primary factors influencing the performance gap are the lower quality and narrower scope of the real dataset, we also recognize that the synthetic-trained model faces challenges in generalizing to real-world scenarios. However, we intentionally embraced this challenge, as we believe that deep learning advancements can progressively close this gap, even in the presence of sensor-induced limitations.
> 4. We do not get what you mean by out-of-distribution data. If you mean other objects, that is not the case. We are doing instance-level 6D pose estimation, also called Model-based 6D detection of seen objects([1]). The models trained on our dataset, widely used in this field, are not zero-shot. If you mean different positions and backgrounds, we assure that each image is completely randomly generated in this sense. In addition, the real dataset is acquired with different objects and positions, backgrounds and lights.
> [1] Bauer, D., et al. (2024). Challenges for monocular 6d object pose estimation in robotics. IEEE Transactions on Robotics.

---

### Official Review · Reviewer_qKsX · 2024-11-03

**Soundness:** 3
**Presentation:** 3
**Contribution:** 1
**Rating:** 3
**Confidence:** 4

**Summary:**

This paper introduces a new object dataset tailored to industrial scenarios. The dataset is generated using a physically-based renderer (PBR) on 10 distinct objects, considering various materials, lighting conditions, and background textures. The paper also conducts extensive experiments on both simulated and real data to assess the realism of the synthetic dataset.

**Strengths:**

The paper is well-written, clear, and provides a thorough literature review of previous works related to object datasets. The experiments are comprehensive and sufficiently demonstrate the effectiveness of the proposed simulated data in real-world scenarios.

**Weaknesses:**

1. A key concern is the limited novelty of the proposed dataset. While the authors emphasize that the dataset includes cluttered scenes, varied lighting conditions, and diverse object materials, these features are already well-supported by modern PBRs [1,2]. Previous works, such as Omni6D [3], have incorporated similar capabilities in the rendering of object datasets. It would be helpful for the authors to further clarify how their dataset differs from prior work in this domain.

2. The dataset contains only 10 object instances, which is quite limited compared to other object datasets. Industrial environments typically feature a much larger variety of objects. Expanding the dataset to include more object types would improve its diversity and applicability.

3. Although the dataset features cluttered scenes, it appears that only top-down viewpoints are used to capture the images. Incorporating images from different viewpoints would enhance the dataset’s diversity and better reflect real-world industrial scenarios.


[1] https://dlr-rm.github.io/BlenderProc/examples/datasets/bop_challenge/README.html

[2] https://docs.omniverse.nvidia.com/isaacsim/latest/replicator_tutorials/tutorial_replicator_pose_estimation.html

[3] Zhang, Jiyao, et al. "Omni6DPose: A Benchmark and Model for Universal 6D Object Pose Estimation and Tracking." arXiv preprint arXiv:2406.04316 (2024).

**Questions:**

From a practical perspective, is it realistic to find industrial scenes where transparent objects and tools are piled together? And is it common for different tools to be stacked separately?

---

> ### Author Response · Authors · 2024-11-18
>
> Thank you for your suggestions. Regarding the weakness:
> 1. We are completely aware of the existence of BlenderProc [1] as a formidable and useful data generator (we cited it in the paper). However they (referring to blenderproc and other similar software) depend on Blender for the rendering part. Which makes it less straightforward to use with respect to the standalone renderer we developed.
> Additionally, BlenderProc and similar tools lack certain critical features that our data generator provides. For instance, they do not handle reflective and transparent objects, which are essential for our dataset. This limitation is significant since metallic reflective objects are frequently encountered in industrial environments.
> Secondly, our renderer is faster, and easier to use as it is fully tailored towards our target use case. We do not report in this paper a performance comparison between blenderproc and our custom renderer, as these are means to generate a dataset, whereas our contribution focuses on the actual dataset.
> Dataset [2] has a low level of realism: objects are randomly posed on a real image.
> Regarding Omni6D [3] : The dataset is real, not proposed for industrial environments, not highly cluttered, and objects do not overlap with each other. The dataset represents a highly valuable source for category level 6D pose estimation advancements, however, the fact that it is real means that we can not generate a dataset on our own objects and directly use it for robotic industrial applications. We will certainly include it in the updated paper’s related works section for comparison.
>
> 2. Given the industrial scenario, where object CAD models are usually available, we focused on instance-level 6D pose estimation. The dataset is in line with other instance-level 6D pose estimation datasets, such as LineMod (15 objects) or YCB-V (21 objects). The objects were requested by a factory for a specific use case. The dataset is easily extendable to additional objects and category-level 6d pose estimation, thanks to the flexible and easy-to-use data generator, this is one of our future purposes. Moreover, scenes are multi-objects and multi-instances, containing scenes up to 20 objects.
>
> 3. You are right, but we chose the top-down viewpoint for a specific reason: our use case is composed of a robotic arm with a camera applied to it. The robotic arm is supposed to take the picture from a top-down view, and then predict the pose for a successful grasping. This is the robotic industrial application we are focusing on. Nevertheless, we can easily create images with variations of x and y axes, if requested by the application. We remark that the z axis is varied in a range between 3.25 and 5 meters (Figure 3.b), given that we consider that the plane could vary in height.
>
> Questions:
> This dataset is the evolution of a project directly committed by an industry. The industry proposed the object models and materials.
> The goal of this article and dataset is precisely to automate industrial scenarios that currently rely on controlled environments, where objects are placed by humans in specific positions pre-communicated to the robot. Currently, robots lack the flexibility to interpret scenes with grouped objects independently. This problem highlights the need to make robots sufficiently intelligent to autonomously understand where objects are located and how to grasp them, even when objects are stacked or involve challenging materials. Furthermore, creating multi-object scenes with diverse materials adds value, as we see it. Training a pipeline with multi-object scenes, if successful, not only prepares it for single-object and multi-instance scenes but potentially enhances its performance in these simpler setups as well. About ‘industry is the most controllable environment’, we claim that industry is going into the automation way and to do that it needs flexible and adaptable robots. Currently, they are far from complete automation. There still exists some repetitive work which only a human can do.

---

### Note · Authors · 2024-12-12

**Comment:**

Given the low rating and the inability to engage in a productive discussion with the reviewers, who did not respond to our messages, we have decided to withdraw the paper.

**Withdrawal Confirmation:**

I have read and agree with the venue's withdrawal policy on behalf of myself and my co-authors.